# Surgical Resection in Colorectal Liver Metastasis: An Umbrella Review

**DOI:** 10.3390/cancers16101849

**Published:** 2024-05-12

**Authors:** Martina Milazzo, Letizia Todeschini, Miriam Caimano, Amelia Mattia, Luca Cristin, Alessandro Martinino, Giuseppe Bianco, Gabriele Spoletini, Francesco Giovinazzo

**Affiliations:** 1Department of Surgery, UpperGI Division Surgery, University of Verona, 37129 Verona, Italy; 2Faculty of Medicine and Surgery, University of Verona, 37129 Verona, Italy; 3Department of Surgery, Fondazione Policlinico Universitario Agostino Gemelli IRCCS, 00136 Rome, Italy; 4Department of Surgery, Duke University, Durham, NC 27708, USA; 5School of Medicine, UniCamillus-Saint Camillus International University of Health Sciences, 00131 Rome, Italy; 6Department of Surgery, Saint Camillus Hospital, 31100 Treviso, Italy

**Keywords:** colorectal liver metastasis, simultaneous colorectal liver metastasis, surgical resection, survival, umbrella review

## Abstract

**Simple Summary:**

Colorectal cancer is a leading cause of death worldwide, with a significant number of patients presenting with liver metastases at diagnosis. The best treatment currently involves the surgical removal of these metastases, yet the optimal timing and method for this surgery are still debated among experts. Our analysis revealed no significant difference in overall survival and disease-free survival between a simultaneous surgery approach and a “bowel-first” approach, while a better 5-year overall survival (5-OS) was found when comparing the former to a “liver-first” approach. However, simultaneous surgeries are associated with a higher risk of perioperative mortality, suggesting that staged surgery might be more beneficial for high-risk patients. This emphasizes the need for personalized treatment plans. Our findings also suggest that anatomic and non-anatomic resections offer similar long-term survival benefits, highlighting the potential for more conservative surgeries to preserve liver function without compromising the effectiveness of cancer treatment.

**Abstract:**

Surgical resection is the gold standard for treating synchronous colorectal liver metastases (CRLM). The resection of the primary tumor and metastatic lesions can follow different sequences: “simultaneous”, “bowel-first”, and “liver-first”. Conservative approaches, such as parenchymal-sparing surgery and segmentectomy, may serve as alternatives to major hepatectomy. A comprehensive search of Medline, Epistemonikos, Scopus, and the Cochrane Library was conducted. Studies evaluating patients who underwent surgery for CRLM and reported survival results were included. Other secondary outcomes were analyzed, including disease-free survival, perioperative complications and mortality, and recurrence rates. Quality assessment was performed using the AMSTAR-2 method. No significant differences in overall survival, disease-free survival, and secondary outcomes were observed when comparing simultaneous to “bowel-first” resections, despite a higher rate of perioperative mortality in the former group. The 5-year OS was significantly higher for simultaneous resection compared to “liver-first” resection. No significant differences in OS and DFS were noted when comparing “liver-first” to “bowel-first” resection, or anatomic to non-anatomic resection. Our umbrella review validates simultaneous surgery as an effective oncological approach for treating SCRLM, though the increased risk of perioperative morbidity highlights the importance of selecting suitable patients. Non-anatomic resections might be favored to preserve liver function and enable future surgical interventions.

## 1. Introduction

Colorectal cancer (CRC) is the third most common cancer in the world and the second most common cause of cancer mortality [1,2]. Approximately 15–30% of patients with colorectal cancer have synchronous liver metastases (SCRLM) at the time of initial diagnosis [3,4,5]. Surgical liver resection is the gold standard in treating CRLM for curative intent treatments in Europe, North America, and Asia [3,6,7]. Achieving a complete resection of hepatic metastasis has been demonstrated to improve survival, resulting in an increase in the five-year survival rate of up to 40% [8]. Surgical resection can be achieved either by performing a major hepatectomy or by using a more conservative approach, such as parenchymal-sparing resection or segmentectomy. There are three main surgical strategies for managing hepatic metastasis, differentiated by the sequence and timing of resections for the primary and metastatic tumors. The simultaneous approach entails resecting both the liver and colon tumors within the same hospital stay. Conversely, the staged approach involves two distinct surgical events: the traditional staged resection typically starts with the removal of the colon cancer, followed by a separate liver surgery during a subsequent hospital admission; on the other hand, the “liver-first” approach initially focuses on performing hepatic surgery, followed by colon surgery. Each strategy aims to optimize outcomes by considering the most effective sequence for tumor removal. However, the current data do not decisively endorse any of these methodologies, and there is still a lack of international consensus regarding the preferred surgical treatment [3,6,7,9].

The present umbrella review aims to systematically evaluate and synthesize the existing evidence from multiple systematic reviews and meta-analyses comparing the different surgical approaches and their outcomes in the treatment of SCRLM.

## 2. Materials and Methods

This systematic review was conducted and reported in accordance with the Preferred Reporting Items for Systematic Reviews and Meta-Analyses (PRISMA) guidance [10]. The review protocol was registered with the PROSPERO international prospective register of systematic reviews (CRD42024525149).

### 2.1. Search Strategy

A computerized search of the Medline, Epistemonikos, and Scopus databases, and the Cochrane Library, was conducted. Articles published from the time of inception to 5 September 2023 were included. An advanced search was conducted using keywords and MeSH terms (“colorectal neoplasms”, “liver neoplasms”, “liver metastasis”, “treatment”) and specifying for publication type (“systematic review”, “meta-analysis”). These terms were linked by operators “AND” and “OR”. Reference lists of all obtained and relevant articles were manually screened and cross-referenced to identify any additional studies by two independent authors (L.T., M.C.). Only articles that evaluated surgical management of SCRLM were selected. All published meta-analyses in the English language were included.

### 2.2. Study Selection

The search results were imported into the research collaboration software Rayyan (https://new.rayyan.ai, accessed on 5 September 2023) [11]. M.M., L.T, A.M (Amelia Mattia), and L.C. independently screened the meta-analysis by reviewing titles and abstracts. Only articles involving subjects diagnosed with CRLM, who were older than 18 years and underwent surgical resection of metastases, were included. Furthermore, included articles had to report survival analysis results in terms of hazard ratio (HR), odds ratio (OR), or relative risk (RR). Only articles in the English language were screened. Conflicts were resolved through discussion with a fifth author A.M. (Alessandro Martinino) and a complete agreement was reached. Full-text versions of the articles included after title/abstract screening were obtained and reviewed by the same reviewers.

### 2.3. Data Extraction and Statistical Analysis

Data encompassing authors, publication year, study types, and the number of studies analyzed were extracted by M.M., L.T., A.M. (Amelia Mattia), and L.C. for data analysis. Data on the cohorts of patients involved, including study arms and the number of patients for each, were also collected. Additionally, a thorough characterization of the disease state within the included patient cohorts was undertaken to ensure a more refined stratification of the results. Data concerning survival outcomes associated with the chosen surgical strategy were collected as they represent the primary endpoint of our study. Furthermore, other secondary endpoints were extracted to comprehensively assess the benefits of each surgical approach, including disease-free survival (DFS), perioperative mortality, postoperative complications, and rates of intrahepatic recurrence. Pooled outcome measures with 95% confidence interval values (95% CI), statistical heterogeneity, and publication bias were also calculated.

### 2.4. Data Analysis

A qualitative summary of the results, using text and tables, was performed. We combined trials from previously published meta-analyses into updated meta-analyses, after removing any duplicates. To provide homogenous and more comparable metrics, we recalculated the summary treatment estimates for each meta-analysis. Effect size for dichotomous outcomes was expressed as RR and OR with their 95% of confidence interval (CI). HR with standard error (SE) were used for time-to-event outcomes. For continuous outcomes, the standardized mean difference (SMD) with 95% CI was used. A *p*-value < 0.05 (two-tailed) was considered to indicate statistical significance. We planned the meta-analysis if there were 2 or more studies with the same outcome. Heterogeneity was assessed using the I2 statistic. I2 values greater than 50% suggest significant heterogeneity; therefore, a random-effect meta-analysis was performed or, otherwise a fixed-effect model. All statistical analyses were performed with RevMan 5, version 5.4.

### 2.5. Quality Assessment

The internal validity of the meta-analyses was assessed by the Assessment of Multiple Systematic Reviews 2 (AMSTAR-2) method. L.T. and M.C. completed the 11-item AMSTAR proforma for all included meta-analyses, and discrepancies were discussed to reach a consensus with a third author (A.M.). Finally, studies were classified based on the level of quality through the online tool calculator.

## 3. Results

### 3.1. Study Selection

A total of 1214 potentially relevant articles were identified using the search strategy described in the Methods section. In total, 618 duplicate records were removed, leaving 596 records available for screening. A total of 561 articles were excluded after abstract reviewing due to inappropriate topic relevance, absence of a meta-analysis, and being in a language other than English. Among the remaining 35 articles, 2 were excluded due to the full text not being available, and another 16 were excluded by examining the full texts because of the absence of survival analysis, comparative analysis, or insufficient data. Finally, 17 meta-analyses met the inclusion criteria and were included in the present umbrella review (Table 1). A summary of the results of the systematic search is shown in the PRISMA 2020 flow diagram (Figure 1).

### 3.2. Quality Assessment Results

Quality levels of the meta-analyses included are reported in the last column of Table 1. Among the 17 meta-analyses included in our umbrella review, all but one demonstrated a critically low quality according to the AMSTAR-2 evaluation. The remaining meta-analysis was assessed as being of low quality [18]. The main common reasons for such low quality were the absence of a registered protocol, the lack of a risk of bias (RoB) assessment, and a poor characterization of the studied populations.

### 3.3. Study Characteristics

A total of 17 meta-analyses were included into our study (Table 1). These publications spanned from 2008 to 2023, collecting data from retrospective and prospective studies. The included patients all had a diagnosis of CRLM or SCRLM. As per the inclusion criteria, all the meta-analyses included in our study examined survival outcomes, which were reported as OS, 3-year OS, and 5-year OS. A wide range of secondary outcomes were investigated: DFS, 3-year DFS, 5-year DFS, perioperative mortality, postoperative complications, recurrence, new intrahepatic recurrence, marginal recurrence, and distant recurrence.

The analysis of the study population revealed the inclusion of both minor and major resections (≥3 segments), as well as both right- and left-sided colon cancer. However, the type of technical approach chosen, meaning open or minimally invasive, was not specified. Studies did not provide details on the modality and timing of follow-up.

### 3.4. Simultaneous Resection vs. Classic Staged (“Bowel-First”) Resection

Twelve meta-analyses compared the simultaneous resection of colorectal cancer and SCRLM with a traditional staged resection approach, where the primary colonic tumor is resected first [4,12,13,15,16,17,18,19,21,23,25,26]. Survival outcomes were reported either in terms of OS or 5-year OS. Patients who underwent simultaneous surgery had comparable survival compared to the staged resection group (HR 0.96 95% CI 0.88–1.05; *p* = 0.40; I^2^ 0%) (Table 2). Similar results were obtained when comparing the differences in 5-year OS between the two surgical approaches (OR 1.05 95% CI 0.86–1.27; *p* = 0.64; I^2^ 70%). DFS and 5-year DFS were also assessed. The pooled meta-analysis showed no significant difference between simultaneous and staged resections in both DFS (HR 0.95, 95% CI 0.67–1.35; *p* = 0.79; I^2^ 72%) and 5-year DFS (OR 0.94 95% CI 0.60–1.47 *p* = 0.78; I^2^ 47%).

Additional secondary outcomes within this set of meta-analyses were accessible and subjected to our analysis (Table 2). Operative time (in min) was assessed in four meta-analyses, and a statistically significant shorter time was observed in the simultaneous surgical strategy (SMD: −0.58; CI [−1.10, −0.07], *p*-value = 0.03, I2 96%). Postoperative complications, including gastrointestinal (ileus, anastomotic leak, abdominal collection), hepatic (bile leak, hepatic insufficiency/failure, subphrenic/perihepatic abscess, bile leakage), and general complications (wound infection, pulmonary and cardiac disease), were assessed. No distinction was made on Clavien–Dindo classification in most of the studies. The postoperative complications rate was comparable between the two groups (RR 0.98; CI 0.90–1.07; *p* = 0.71; I^2^ 50%). Intraoperative blood loss (in mL) among patients undergoing simultaneous or staged surgery did not reveal a statistically significant difference (SMD −0.35; 95% CI −0.92–0.23; *p* = 0.24; I^2^ 97%). The perioperative blood transfusion rate was reported in one meta-analysis, although this difference did not reach statistical significance (RR 1.11; CI 0.96–1.30; *p* = 0.17; I^2^ 40%). Examination of perioperative mortality rates unveiled a statistically significant increase for the simultaneous surgical approach when juxtaposed with the bowel-first approach (RR 1.88; 95% CI 1.33–2.65; *p* < 0.001; I^2^ 11%). Analysis of the duration of hospitalization following simultaneous or staged resection demonstrated a statistically significant reduction in length of stay within the first group (SMD −1.08; 95% CI −1.55–0.60; *p* < 0.001; I^2^ 98%). Finally, recurrence after resection was found to be comparable between the simultaneous resection and the staged resection groups, with no statistically significant difference reached (RR 1.04; CI 0.91 −1.19; *p* = 0.56; I^2^ 62%).

### 3.5. Simultaneous Resection vs. “Liver-First” Surgery

Two meta-analyses compared simultaneous primary tumor and SCLRM resection to a liver-first staged approach [19,23]. Our meta-analysis demonstrated a noteworthy difference in 5-year OS rates between patients undergoing simultaneous resection and those undergoing hepatic surgery first (Table 2). Specifically, patients in the simultaneous resection group exhibited a better 5-year OS compared to those in the liver-first surgery group (OR 0.47 95% CI 0.25–0.90 *p*= 0.02). No interstudy heterogeneity was present (I^2^ 0%).

### 3.6. Liver-First Surgery vs. Bowel-First Surgery

The liver-first approach and bowel-first approach were compared in three meta-analyses [19,23,24]. Five-year OS (OR 1.15; CI 0.93 −1.42; *p* = 0.19; I^2^ 18%), and five-year DFS (OR 1.01; CI 0.72–1.42; *p* = 0.96; I^2^ 0%) did not reveal a significant advantage in either group (Table 2).

### 3.7. Anatomic vs. Non-Anatomic Surgery

Four studies were reviewed to compare the outcomes of anatomic versus non-anatomic surgical resection in the treatment of liver metastases from colon cancer, focusing on 3-year and 5-year OS and DFS [14,20,22,27]. However, these studies lacked detailed descriptions regarding follow-up methods and timing. Anatomic resection was defined as the systematic removal of liver parenchyma corresponding to one or more of Couinaud’s segments, delineated by portal venous branches. In contrast, non-anatomic resection referred to a parenchyma-sparing hepatectomy that achieved an adequate resection margin. Our comparative analysis across these studies revealed no significant difference either in OS (HR 1.04, 95% CI 0.98–1.11, *p*-value = 0.16, I^2^ 31%) or in DFS (HR 1.08, CI 0.97–1.20, *p*-value = 0.17, I^2^ 59%) (Table 3).

## 4. Discussion

Between 25% and 50% of patients with colorectal cancer will develop colorectal liver metastases (CRLM) at some point during their disease progression. Almost half of the patients diagnosed with colorectal liver metastasis present with metastases that are confined to the liver [28]. Surgical resection is considered the gold standard treatment for patients with resectable colorectal liver metastases as it correlates with higher survival rates compared to systemic oncologic treatments in various cohorts [29,30,31]. Even in cases where colorectal liver metastases are initially deemed non-resectable, a combined treatment strategy that pairs surgical intervention with systemic therapy may still be contemplated. This approach can be applied to both downstage the metastases and potentially convert them to a resectable status, paving the way for surgical removal [3,6,7]. Current treatment modalities for SCLM encompass a variety of approaches, including surgical resection, radiofrequency ablation, cryosurgery, hepatic arterial infusion, and systemic chemotherapy. The effectiveness of surgical treatment of liver metastases from colorectal cancer is well-supported by evidence in the literature, with liver resection leading to five-year survival rates that exceed 50% [32]. Various strategies exist regarding both the timing and extent of surgery. Regarding timing, the choice between simultaneous or staged resection for SCLM is subject to ongoing debate, with no established consensus on the criteria for surgery or the optimal timing for such interventions [3,6,7,9].

In our umbrella review, we analyzed long-term OS and DFS across various surgical strategies for treating SCRLM, specifically examining the “bowel-first”, “liver-first”, and “simultaneous” resection approaches. Our findings revealed that there were no significant differences in OS and DFS when comparing simultaneous resections to “bowel-first” resections. Furthermore, when comparing these two surgical strategies, we found them to be similar in terms of postoperative outcomes, including complications, recurrence rates, blood loss, and the need for perioperative blood transfusion. Shorter hospitalization and operative times were observed in the simultaneous surgical approach group. However, a significantly higher rate of perioperative mortality was associated with the simultaneous resection group. This observation suggests that a staged surgical approach might offer advantages, particularly for patients who may have a higher risk profile. It underscores the significance of personalized treatment planning in managing SCRLM, in line with the recommendations of major guidelines. Future research should compare minor and major hepatectomies in the simultaneous approach to better understand the relationship between the extent of hepatic surgery and its impact on mortality and morbidity rates. Additionally, further studies should assess perioperative morbidity by evaluating the severity of complications using established international classification systems, such as the Clavien–Dindo scale, to ensure a more standardized and interpretable analysis. Finally, it would be beneficial to perform an additional analysis specifically targeting the surgical approach to ascertain the impact of minimally invasive surgery (MIS) and laparoscopic surgery (LS) on the duration of hospital stays in comparison to the traditional open approach, an aspect that could not be covered in our current analysis. Several studies have posited that both MIS and LS are linked to shorter hospitalizations relative to open surgery [33,34,35,36]. However, there remains a notable gap in the literature regarding systematic reviews or meta-analyses that compare MIS and LS directly, particularly in the context of combined colon and liver surgeries. Addressing this gap could provide more precise insights into the benefits and drawbacks of each surgical approach in the management of complex surgical cases. Perioperative mortality rates were not available. Some studies suggest that major hepatectomy in “simultaneous surgery” should be performed only in young patients without rectal surgery [37,38]. Future analyses should establish selection criteria based on patient characteristics (age, comorbidities), primitive tumor (site, molecular biology, grading), and metastasis features (number, size, right- or left-liver) to determine which patients would truly benefit from a simultaneous approach.

Similar 5-year OS outcomes were observed when comparing “liver-first” to “bowel-first” surgeries, while a notably higher 5-year OS was observed when opting for a simultaneous resection of the lesions over a “liver-first” approach, indicating a clear advantage for simultaneous resection.

Our investigation sought to discern the impact of anatomic (segmentectomy) versus non-anatomic (parenchyma-sparing) resections on the treatment outcomes for SCRLM, with a particular focus on OS and DFS at both 3 and 5 years. The comparative analysis of these surgical strategies revealed no significant differences in OS and DFS outcomes. This finding adds to the ongoing debate regarding the optimal approach to hepatic resection in the case of CRLM and the effectiveness of “sparing hepatic parenchyma” both for oncological margins and to improve residual liver function [39]. Indeed, while some studies have indicated improved survival rates with segmentectomy as the preferred intervention, other research has shown no significant disparity in long-term outcomes between anatomic and non-anatomic resections [22,40]. Notably, non-anatomic resections have the advantage of preserving sufficient liver parenchyma, potentially facilitating future resections if necessary. Given the similar survival rates, our findings suggest that non-anatomic parenchymal-preserving resection might be especially preferable when further surgical intervention is anticipated.

Some limitations must be addressed when analyzing the results of our meta-analysis. The quality of the included meta-analyses, assessed using AMSTAR-2 criteria, was found to be generally critically low, necessitating prudence in the interpretation of their findings. Additionally, our analysis could not take into consideration the number and the size of the metastases and the primary tumor, as well as the primary tumor characteristics (e.g., localization, molecular biology, grading), as the available data were too scarce. Furthermore, the criteria for patient selection remained vague due to the lack of a clear definition for “synchronous resectable metastasis”. The term “resectable” raised questions: it was uncertain whether it pertained solely to technical feasibility (whether surgical removal was possible) or also encompassed oncological adequacy (achieving a margin-negative, or R0, resection), or both. Additionally, most of the studies neglected to clarify when metastases were identified, nor did they elaborate on the specifics of follow-up in terms of timing and methodology. They also did not specify the decision to use adjuvant or neoadjuvant therapy, which is recognized in current guidelines as a crucial aspect of patient management. The retrospective nature of some studies introduced a selection bias, especially evident in the inconsistent numbers of patients undergoing varying extents of hepatectomy, from minor to major. Our umbrella review spans a wide time frame, which also introduces a potential time-period bias. Over the past decade, cancer treatment has evolved significantly as molecular alterations have been recognized as key drivers of cancer development and progression [41]. During this time, clinical indications and inclusion criteria have also evolved and may vary between the original investigations included in the meta-analyses. Moreover, the introduction of more refined imaging assessments and the use of molecular biology tests to identify specific biomarkers, such as MSI/MSS, Ras mutations, BRAF mutations, and HER2 amplification, have become crucial in determining appropriate management strategies for colon cancer [3,42,43,44,45]. This paradigm shift represents a transition from an “organ-centric” to a “patient-centric” approach in selecting therapeutic strategies, underscoring the vital role of multidisciplinary teams [3]. These teams function as collaborative roundtables where healthcare professionals regularly discuss each colon cancer case in detail. Such an approach allows for careful selection of both patients and surgical strategies (including timing and technique), integrating molecular biomarkers, imaging features, and tumor clinicopathologic factors with patient-specific characteristics for optimal outcomes. Future studies should consider these advancements in management to standardize patient cohorts.

## 5. Conclusions

Our umbrella review confirms the role of simultaneous surgery as an oncologically safe and effective treatment for SCRLM, though its higher risk of perioperative morbidity underscores the need for careful selection of fit patients. In terms of the extent of surgery, parenchyma-preserving resections may be preferred to maintain liver function and facilitate future surgeries, without compromising long-term outcomes. Further investigations should also strive for more precisely defined patient cohorts to minimize the impact of selection biases and indeed account for the variability that may impact the effectiveness of each technique. Such research would provide a more precise and more definitive comparison of the different approaches, thereby offering more reliable guidance for making clinical decisions.

## Figures and Tables

**Figure 1 cancers-16-01849-f001:**
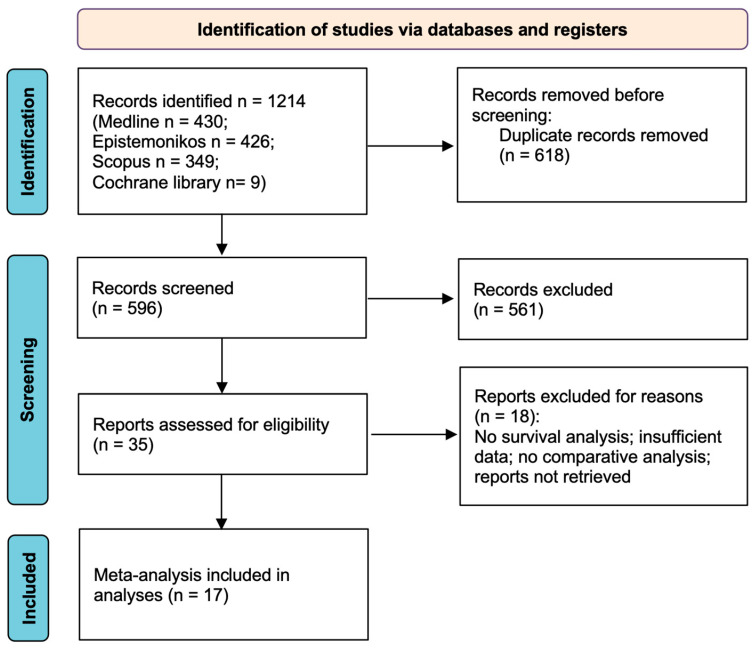
PRISMA flow diagram.

**Table 1 cancers-16-01849-t001:** Study characteristics.

Author, Year	Primary Studies Design (n)	Purpose	N° of Patients (n)	Condition	Comparison	Outcome (n Studies)	Metrics [95% CI]	*p*-Value	I^2^ (%)	Quality
Hillingsø JG, 2008 [12]	R (16)	To determine the level of evidence for recommendations of a treatment strategy of synchronous liver metastases from colorectal cancer	1650	Synchronous CRLM	Simultaneous vs. staged surgery	5-year OS (11)	OR: 0.90 [0.66,1.24]	0.14	24	Critically low
Chen J, 2011 [13]	R (14)	To compare outcomes between simultaneous resection and staged resection from all published comparative studies in the literature	2204	Synchronous CRLM	Simultaneous vs. staged surgery	5-year OS (10)	OR: 1.14 [0.86,1.50]	0.37	0	Critically low
Sui CJ, 2012 [14]	R (7)	To compare anatomic resection versus non-anatomic resection for colorectal liver metastases with respect to perioperative and oncological outcomes	1662	CRLM	Anatomic vs. non-anatomic resection	5-year DFS (2)	OR: 1.27 [0.66,2.42]	0.47	61.5	Critically low
Li ZQ, 2013 [15]	R (19)	To identify the optimal timing of surgical resection for colon and liver in CRLM patients	2724	Synchronous CRLM	Simultaneous vs. staged surgery	5-year OS (13)	OR: 1.12 [0.88,1.42]	0.38	0	Critically low
						5-year DFS (3)	OR: 0.61 [0.33,1.13]	0.12	0	
Yin Z, 2013 [16]	R (17)	To define the safety and efficacy of simultaneous versus delayed resection of the colon and liver in CRLM patients	2880	Synchronous CRLM	Simultaneous vs. staged surgery	DFS (4)	HR: 1.04 [0.76,1.43]	0.79	53	Critically low
						OS (10)	HR: 0.96 [0.81,1.14]	0.64	0	
Bijukchhe SM, 2014 [17]	R (20)	To evaluate the efficacy between simultaneous resection and staged resection in patients with CRLM	3194	Synchronous CRLM	Simultaneous vs. staged surgery	5-year OS (12)	OR: 1.08 [0.84,1.38]	0.54	0	Critically low
						5-year DFS (4)	OR: 0.60 [0.34,1.04]	0.07	0	
Feng Q, 2014 [18]	R (22)	To compare simultaneous resection with staged strategy in colorectal cancer for patients with synchronous liver metastases patients, balancing baseline characteristics	4494	Synchronous CRLM	Simultaneous vs. staged surgery	OS (15)	HR: 0.96 [0.86,1.08]	0.50	0	Low
						DFS (6)	HR: 0.97 [0.64,1.47]	0.87	76	
Kelly ME, 2014 [19]	R (18)	To compare the short- and long-term outcomes for the classical colorectal-first, the liver-first, and the combined resections approaches	3605	Synchronous CRLM	Simultaneous vs. liver first	5-year OS (3)	OR: 0.78 [0.44,1.40]	0.410	0	Critically low
					Bowel first vs. liver first	5-year OS (4)	OR: 0.85 [0.59,1.22]	0.374	69.8	
					Bowel first vs. simultaneous	5-year OS (14)	OR: 0.99 [0.83,1.19]	0.927	5.3	
Tang H, 2016 [20]	R (21)	To compare the efficacy of anatomic resection procedure and non-anatomic procedure for CRLM	5207	CRLM	Anatomic vs. non-anatomic resection	OS (12)	HR: 1.06 [0.95,1.18]	0.18	27.3	Critically low
						DFS (5)	HR: 1.11 [0.99,1.24]	0.76	0	
Gavriilidis P, 2018 [21]	R (30)	To assess outcomes between patients undergoing simultaneous or delayed hepatectomy for synchronous colorectal liver metastases	5300	Synchronous CRLM	Simultaneous vs. staged surgery	OS (17)	HR: 0.97 [0.88,1.08]	0.601	0	Critically low
Deng G, 2019 [22]	R (17) P (1)	To assess the safety and efficacy of parenchymal-sparing hepatectomy as a treatment of colorectal liver metastases	7081	CRLM	Anatomic vs. non-anatomic resection	OS (16)	HR: 1.01 [0.94,1.08]	0.82	0	Critically low
						DFS (11)	HR: 1.00 [0.94,1.07]	0.92	0	
Ghiasloo M, 2020 [23]	R (43) P (1)	To compare the bowel-first approach, simultaneous resection, and the liver-first approach for patients with colorectal cancer with synchronous liver metastases	10,848	Synchronous CRLM	Liver first vs. bowel first	5-year DFS	OR: 0.57 [0.16,2.06]	0.39	91	Critically low
						5-year OS	OR: 0.88 [0.54,1.44]	0.61	44	
					Simultaneous vs. bowel first	5-year DFS	OR: 1.07 [0.71,1.62]	0.74	31	
						5-year OS	OR: 0.90 [0.75,1.08]	0.26	14	
					Simultaneous vs. Liver first	5-year OS	OR: 0.47 [0.25,0.90]	0.02	0	
Hajibandeh S, 2020 [4]	R (41)	To evaluate the comparative outcomes and clinical characteristics of simultaneous and staged colorectal and hepatic resections for colorectal cancer with synchronous hepatic metastases	12,081	Synchronous CRLM	Simultaneous vs. staged surgery	5-year OS (23)	OR: 0.88 [0.73,1.07]	0.19	45	Critically low
Magouliotis DE, 2020 [24]	R (8) P (2)	To compare the perioperative outcomes of liver-first and classical strategy for the management of synchronous colorectal liver metastases	3656	Synchronous CRLM	Bowel first vs. Liver first	5-year DFS (4)	OR: 1.25 [0.78,1.99]	0.36	9	Critically low
						5-year OS (6)	OR: 1.23 [0.75,2.03]	0.42	66	
Liu J, 2022 [25]	R (17) P (1)	To compare the difference between patients with CRLM who underwent simultaneous resection and staged resection, especially during major hepatectomy (≥3 liver segments)	5101	Synchronous CRLM	Simultaneous vs. staged surgery	5-year mortality (13)	OR: 1.08 [0.93,1.25]	0.337	33.5	Critically low
Wang SH, 2022 [26]	R (22) RCT (1)	To evaluate the safety and long-term prognoses of simultaneous and staged resection of CRLM	4862	Synchronous CRLM	Simultaneous vs. staged surgery	5-year DFS (4)	HR: 1.26 [0.96,1.66]	0.098	18.1	Critically low
						5-year OS (10)	HR: 1.13 [0.95,1.34]	0.164	34.6	
Wang K, 2023 [27]	R (22)	To evaluate the safety and efficacy of parenchymal-sparing resection over anatomic resection for colorectal liver metastases	7228	CRLM	Anatomic vs. non-anatomic resection	OS (22)	HR: 1.08 [0.95,1.22]	0.245	49.3	Critically low
						DFS (14)	HR: 0.84 [0.41,1.73]	0.259	75.1	

Abbreviations: R: retrospective study; P: prospective study; RFA: radiofrequency ablation; LR: liver resection; MWA: microwave ablation; CLRM: colorectal liver metastasis; CI: confidence interval; OS: overall survival; DFS: disease-free survival; HR: hazard ratio; OR: odds ratio; RR: relative risk.

**Table 2 cancers-16-01849-t002:** Meta-analyses results: comparison of simultaneous, “bowel-first”, and “liver-first” resection.

Parameter	Meta-Analyses Included (n)	Primary Studies Included (n)	Patients in the First Group (n)	Patients in the Second Group (n)	Metrics [95% CI] Recalculated	*p*-Value	I^2^ (%)
*Simultaneous resection vs. “bowel-first” resection*							
OS	4	23	1750	2242	HR: 0.96 [0.88, 1.05]	0.40	0
DFS	2	7	544	397	HR: 0.95 [0.67, 1.35]	0.79	72
5-y OS	9	38	3774	6211	OR: 1.05 [0.86, 1.27]	0.64	70
5-y DFS	4	5	245	275	OR: 0.94 [0.60, 1.47]	0.78	47
Perioperative mortality	10	21	2259	3842	RR: 1.88 [1.33, 2.65]	<0.001	11
Postoperative complications	9	44	4486	5415	RR: 0.98 [0.90, 1.07]	0.71	50
Recurrence	3	14	1318	1455	RR: 1.04 [0.91, 1.19]	0.56	62
Blood loss, ml	5	9	642	1261	SMD: −0.35 [−0.92, 0.23]	0.24	97
Perioperative blood transfusion	1	9	616	1213	RR: 1.11 [0.96, 1.30]	0.17	40
Hospitalization, days	8	26	3233	3137	SMD: −1.08 [−1.55, −0.60]	<0.001	98
Operative time, min	4	13	923	950	SMD: −0.58 [−1.10, −0.07]	0.03	96
*Simultaneous resection vs. “liver-first” resection*							
5-y OS	2	3	88	86	OR: 0.47 [0.25, 0.90]	0.02	0
*“Liver-first” vs. “bowel-first” resection*							
5-y OS	3	8	525	1783	OR: 1.15 [0.93, 1.42]	0.19	18
5-y DFS	3	5	232	1305	OR: 1.01 [0.72, 1.42]	0.96	0

Abbreviations: CI: confidence interval; OS: overall survival; DFS: disease-free survival; HR: hazard ratio; OR: odds ratio; RR: relative risk; SMD: standardized mean difference.

**Table 3 cancers-16-01849-t003:** Meta-analyses results: comparison of anatomic and non-anatomic resection.

Parameter	Meta-Analyses Included (n)	Primary Studies Included (n)	Patients Undergoing Anatomic Resection (n)	Patients Undergoing Non-Anatomic Resection (n)	Metrics [95% CI] Recalculated	*p*-Value	I^2^ (%)
**OS**	4	27	3368	4399	HR: 1.04 [0.98, 1.11]	0.16	31
**DFS**	4	17	2660	3763	HR: 1.08 [0.97, 1.20]	0.17	59

Abbreviations: CI: confidence interval; OS: overall survival; DFS: disease-free survival; HR: hazard ratio; RR: relative risk.

## Data Availability

Data are contained within the article.

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
