# Peer review of "Surgical Resection in Colorectal Liver Metastasis: An Umbrella Review"

_cancers, 2024, doi:10.3390/cancers16101849_

Round 1

Reviewer 1 Report

Comments and Suggestions for Authors

The reviewed compared different surgical approaches for treating synchronous colorectal liver metastases, including simultaneous, bowel-first, and liver-first resections. It analyzed survival outcomes, perioperative complications, and recurrence rates across these approaches. The findings suggest that simultaneous resection is an effective oncological strategy for SCRLM, with higher 5-year overall survival compared to liver-first resections. However, simultaneous resections carry a higher risk of perioperative mortality, emphasizing the need for careful patient selection. This review provided useful information on the surgical treatment options for colorectal liver metastasis, some revisions are recommended. 

1. It is suggested that the authors could offer insights and recommendations based on their comprehensive literature analysis regarding which patients may benefit most from the different surgical strategies for treating synchronous colorectal liver metastases: "simultaneous," "bowel-first," and "liver-first" resections.

2. Key factors that influence the outcomes of these surgical approaches need to be discussed. For instance, the patient-specific factors such as the extent of liver involvement, tumor size, location, comorbidities, and overall health status. These factors can help determine the optimal timing and sequence of surgeries for each patient.

3. Considering the varying rates of perioperative complications and mortality associated with each approach, the authors can provide nuanced recommendations tailored to different patient profiles. For instance, patients with lower perioperative risk profiles may benefit from simultaneous resections due to its potential for improved long-term survival, whereas those with higher risk profiles may require a more cautious approach with staged surgeries.

4. Although liver transplantation may not be a primary consideration within the context of the surgical strategies discussed in the review. It would be ideal to discuss the role and indications of liver transplantation comparing the three strategies?

Comments on the Quality of English Language

Minor editing.

Reviewer 2 Report

Comments and Suggestions for Authors

Very nice review, comparing the existing evidence and clearly indicating the gold standard.

The Methods section gives a detailed explanation of the allocated papers and selection of manuscripts further compared.

In the results the techniques are compared and analysed and then discussed. The authors clearly point out that various strategies exist regarding both the timing and extent of surgery and that in terms of timing, the choice between simultaneous or staged resection for SCLM is subject to ongoing debate, with no established consensus on the criteria for surgery or the optimal timing for such interventions. The presented review confirms the role of simultaneous surgery as an oncologically safe and effective treatment for SCRLM, though its higher risk of perioperative morbidity underscores the need for careful selection of fit patients. Certainly an important point, where others in the past have already shown, that experience and exposure of the group matters.

Reviewer 3 Report

Comments and Suggestions for Authors

One of the problems with reviewing the published literature for a 16 year time frame, for what is an ever evolving clinical topic is that this introduces other limitations into the results which the authors need to address in more detail in the Discussion section. The decision making underpinning how patients were managed in 2008 may not be the same in recent years (particularly the last 5)-

1) Nothing is known about whether the patients were being offered adjuvant chemotherapy and hence whether this was impacting on the decision making in some of these studies. 

2) More accurate imaging, including the use of FDG-PET in more recent years may be impacting on the decision making as to which pathway the patients in the more recent studies are being directed down (along with location of the primary tumor and the results of tumor markers-which are only briefly touched on in the Discussion)

3) The lack of information on what performance/medical suitability thresholds the patients needed to meet to undergo either type of surgery is also problematic. Plus, as more experience has been gained with managing older patients through major surgery in the last decade, again this imposes a time factor bias with respect to the published literature

3) There is no mention of MDT decision making (which increasingly is the mode of practice in many places, which does need to be mentioned in the Discussion section)

4) There is no mention/discussion of how these results relate to the published guidelines that have been developed which address the management of metastatic colorectal malignancy (including the role of liver resection, adjuvant chemotherapy and locoregional therapies), for eg the ESMO guidelines (which are Reference number 3 in this particular draft manuscripts Reference list-and is referred to but only in an indirect manner )

Reviewer 4 Report

Comments and Suggestions for Authors

This is a very nice umbrella review on prognostic impact of differen7 strategical approaches in surgery of colorectal liver metastases. The issue is of great importance in view of clinical significance and patient load. The manuscript is well written and adequately discussed. The findings are worth being pusblished. I do not have to state any critical point.

Round 2

Reviewer 3 Report

Comments and Suggestions for Authors

I am happy that all of the reviewers concerns have been adequately addressed via the amendments that have been made to the manuscript